# Microstructure and Corrosion Resistance of Fusion Welding Zone for Duplextubes Welded with Q345R Tube Sheet under Different Welding Currents

Guofu Ou [1,*], Guangwei Qian [1], Haozhe Jin [2,*], Wangping Wu [1,*] and Qianqian Li [1]

1   School of Mechanical Engineering and Rail Transit, Changzhou University, Changzhou 213164, China; 19080706699@smail.cczu.edu.cn (G.Q.); liqianqian@cczu.edu.cn (Q.L.)
2   Institute of Flow-Induced Corrosion, Zhejiang Sci-Tech University, Hangzhou 310018, China
*   Correspondence: ougf@163.com (G.O.); haozhe2007@163.com (H.J.); wwp3.14@163.com (W.W.)

**Abstract:** Duplextubes are widely used in oil and gas storage and transportation, the nuclear industry, and other fields, but the welding quality of metals is an important factor affecting the use of equipment. In order to study the welding quality of S10C steel/Incoloy 825 duplextubes and Q345R tube sheet based on gas tungsten arc welding technology with a filler of ER50-6 carbon steel welding wire, the microstructure and grain size of fusion welding zone of duplextubes and tube sheet under welding currents of 150 A, 160 A, and 170 A were studied by optical microscopy and scanning electron microscopy. At the same time, the corrosion behavior of fusion welding zone after the welding was investigated in 3.5 wt.% NaCl solution by potentiodynamic polarization and electrochemical impedance spectroscopy. The results show that the metallurgical structure of the fusion welding zone was mainly composed of δ-ferrite and retained austenite. The grain size in the fusion welding zone increased with the increase of the welding current. The corrosion resistance of the fusion welding zone welded with a high welding current of 170 A was better than that with low welding currents. However, the pitting corrosion resistance of fusion welding zone with the lowest welding current of 150 A was better than that with high welding currents. This study can provide a preliminary exploration for the manufacture and applicability of duplextubes air coolers.

**Keywords:** duplextubes; welding; microstructure; corrosion

## 1. Introduction

A hydrogenation high-pressure air cooler has the characteristics of high temperature, high pressure, and near hydrogen, which plays an important role in the petroleum refining process [1]. In recent years, with the continuous aggravation of crude oil inferior process, air cooler corrosion failure accidents have occurred frequently, which greatly affects the safe production and operation of the device [2–4]. Therefore, in order to ensure the safe operation of the equipment, carbon steel air coolers are usually upgraded to Incoloy 825 (UNS designation N08825). Incoloy 825 is a solid-solution strengthened alloy based on Ni-Fe-Cr system with additions of Mo, Cu and Ti [5]. In this superalloy, nickel is one of the main alloying elements, providing good resistance to nitridation, carburization, halogenation, and chloride stress-corrosion cracking [6]. Moreover, in combination with molybdenum and copper elements, it providesexcellent resistance to reducing acids. In addition, molybdenum improves resistance to pitting and crevice corrosion and chromium protects the alloy from oxidizing and sulfiding environments by the formation of $Cr_2O_3$. The presence of Ti in the composition of this alloy helps it to maintain its strength and hardness at high temperatures via precipitation of the carbide, nitride, and/or carbonitrides of this element. This alloy is widely used in the chemical and petrochemical industries, oil and gas refining, acid production units, and super-heaters in waste heat incinerators. However, the cost of Incoloy 825 is ten times that ofthe ordinary carbon steel air cooler sincethe

corrosion failure area in the air cooler is prone to be only the tube bundle. The duplextube has good mechanical properties, corrosion resistance and relatively low cost, which has been applied in oil and gas transportation and other fields [7,8]. Using duplextubes instead of carbon steel air cooler tube bundle can not only improve the safety of the air cooler but also take into account the problem of its high manufacturing cost.

A duplextube mainly includes a mechanical composite pipe and metallurgical composite pipe (which is composed of outer base pipe and inner lining pipe with good corrosion resistance). Due to the large difference in the mechanical and corrosion performance of internal and external metal by welding technology, and different coefficients of thermal expansion (it is easy to crack when it is required for welding), the quality of welding seam largely restricts the large-scale promotion and application of duplextubes. Lu et al. [9] conducted tests on the intergranular corrosion resistance, pitting corrosion resistance and stress corrosion resistance of X60/825 duplextubes produced by explosive welding + JCO molding, and found that the corrosion rate of the inner cladding material of the duplextube is equal to that of the raw material. Shui et al. [10] studied the mechanical properties and corrosion resistance of the welded joint of L245NCS/316L duplextubes, and showed that the mechanical properties of the welded joint, such as yield strength and tensile strength, were studied. No cracks appeared in the welded joint in the intergranular corrosion resistance and chloride-stress corrosion. Zhang et al. [11] studied the interfacial structure and mechanical properties of AZ31B/AA6061 composite board by explosive welding method, and showed that the interfacial bonding was good. During annealing process, the magnesium and aluminum diffuse at the interface of composite panels and the intermetallic compound layer was generated. With the increasing of annealing temperature and time, the thickness of compound layer increased obviously, and the elongation of composite board increased obviously by increasing tempering temperature. Lv et al. [7] conducted 20G/316L duplextubes by tungsten inert gas welding technique, and showed that the weld area was divided into four parts: a carbon steel pipe layer, diffusion layer, transition layer, and stainless steel. The welding parameters are reasonable, the mechanical properties of the welded joints are excellent, and there is no evidence of defects. Up to now, the research mostly focuses on the mechanical properties, chemical composition, microstructure, and corrosion resistance of the welds of duplextubes with large diameter used in the fields of oil and gas exploitation and transportation. However, there are few studies on the welding of duplextubes with small inner diameter used in high-risk equipment in petroleum refining and other fields. The high residual stresses, stress corrosion cracking, and metallurgical problems such asforming an unmixed zone, $\delta-$Fe formation in the heat affected zone (HAZ), carbon migration, fusion zone, and hydrogen-induced cracking occur in the weldment [12,13]. These problems must be evaluated before applicationin various industries.

In this study, the duplextubes with base layer of S10C steel and cladding layer of Incoloy 825 alloy were welded with Q345R tube sheet by gas tungsten argon arc welding technology under different welding currents. The microstructure and corrosion behavior of the weld joints under different welding currents were compared and studied.

## 2. Experimental

The weld zone (WZ) of the tube bundle and tube sheet was welded by gas tungsten argon arc welding technology under different welding currents. Figure 1a shows the digital camera photo of welding machine (ZX7-630, Taian Treasure of Electrical Equipment CO., LTD, Taian, China). The material of the tube sheet is Q345R, the specification is $200 \times 200 \times 30$ mm$^3$, and the delivery state is hot-rolled. After testing, the quality and tolerance of the inner and outer surfaces of the tube sheet are qualified and the requirements of welding were met. The heat exchange tube (Shanghai Tianyang Steel Tube Co., Ltd., Shanghai, China) is made of duplextube with a size of $\Phi25 \times 3.15$ mm$^2$. The base material of the heat exchange tube is S10C steel with a thickness of 2.5 mm, and the cladding material is an Incoloy 825 alloy with a thickness of 0.65 mm. The base pipe is connected

with the cladding pipe by means of the pressure melting anchoring bimetal metallurgy technology, as shown in Figure 1b. The formed dupluxtube was heat-treated at temperature of 650–1100 °C. The filler metal material for the welding is ER50-6 carbon steel welding wire with a diameter of 1 mm. The chemical composition of the S10C base pipe, Q345R tube sheet, and ER50-6 filler metal is displayed in Table 1. The three kinds of materials belong to a series of carbon steel, therefore the coefficients of thermal expansion for three kinds of steels are almost matched.

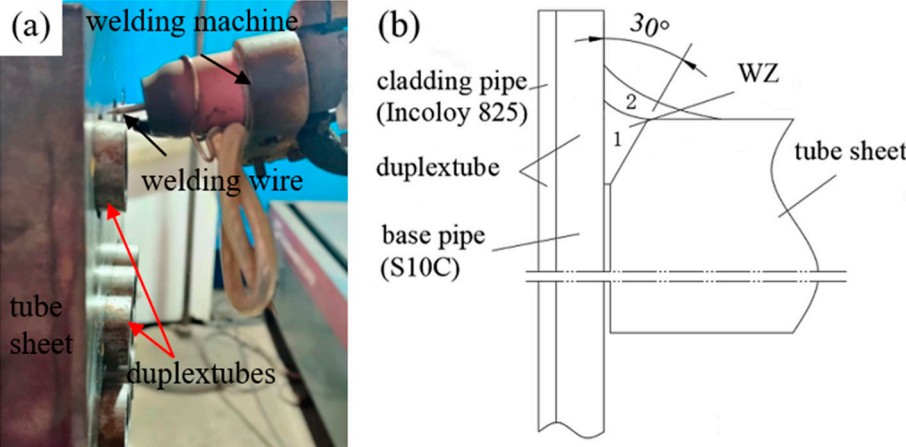

**Figure 1.** Digital camera photo (**a**) of welding machine and schematic diagram (**b**) of welded joints (WZ: weld zone, 1—the first weld seam, 2—the second weld seam).

**Table 1.** Chemical composition of duplextube tubes, plates and filler metals (wt.%).

| Element | Fe | C | Cr | Ni | Mn | Si | P | S | Mo |
|---------|------|-------|-------|-------|------|------|-------|-------|-------|
| S10C | 99.188 | 0.09 | 0.03 | 0.01 | 0.44 | 0.22 | 0.014 | 0.007 | 0.001 |
| Q345R | 97.842 | 0.19 | 0.02 | 0.01 | 1.51 | 0.41 | 0.012 | 0.003 | 0.003 |
| ER50-6 | 97.411 | 0.078 | 0.025 | 0.019 | 1.55 | 0.87 | 0.025 | 0.012 | 0.01 |

The duplextubes and tube sheet are welded by gas tungsten gas argon arc welding technologywith two-pass welding. The welding joints in the form of fillet joint, as shown in Figure 1, were carried out under three different welding currents of 150 A, 160 A, and 170 A, namely samples #1, #2, and #3, respectively. The selection of welding current is determined by empirical value. The welding speed and voltage were set at 2 mm/s and 14 V, respectively. The welding parameters are displayed in Table 2. In order to ensure the welding quality, a V-shaped groove with an angle of 30° was processed on the tube sheet. Before welding, mechanical and chemical cleaning was performed with acetone to remove impurities on the surface of the welding area. The weld input heat (*E*) per unit weld length can be used by the following formula (1) [14]:

$$E = \eta IU/S \tag{1}$$

where *I* represents the welding current (A), *U* represents the welding voltage (V), *S* represents the welding speed (mm/s); $\eta$ is the thermal efficiency coefficient, and the thermal efficiency coefficient of argon arc welding is taken as 0.75 [14], and *E* represents welding heat input (kJ/mm).

**Table 2.** Welding parameters (speed: 2 mm/s, voltage: 14 V).

| Parameters | Current (A) | Weld Heat Input (kJ/mm) |
|:---:|:---:|:---:|
| #1 | 150 | 0.788 |
| #2 | 160 | 0.840 |
| #3 | 170 | 0.893 |

The welded samples were cut from the welded joint by a wire cutting machine, as shown in Figure 2, in order to study microstructure and corrosion resistance of the welded joints. Firstly, we used 400-mesh, 800-mesh, 1200-mesh, and 2000-mesh sandpapers to polish the surface of the samples. Then, these polished samples were immersed and etched in 4% nitric acid alcohol solution for 10 s, and the surface was observed by magnification 500× and 1000× using an optical microscope (VHX-700FC, KEYENCE, Shenzhen, China). The morphology and thickness of the cross-section of the duplextubes were observed by scanning electron microscopy (SEM, FEI CO., JSM-6360LV, Tokyo, Japan). The chemical composition of the sample was determined by X-ray energy-dispersive spectroscopy (EDS, X-act) detector.

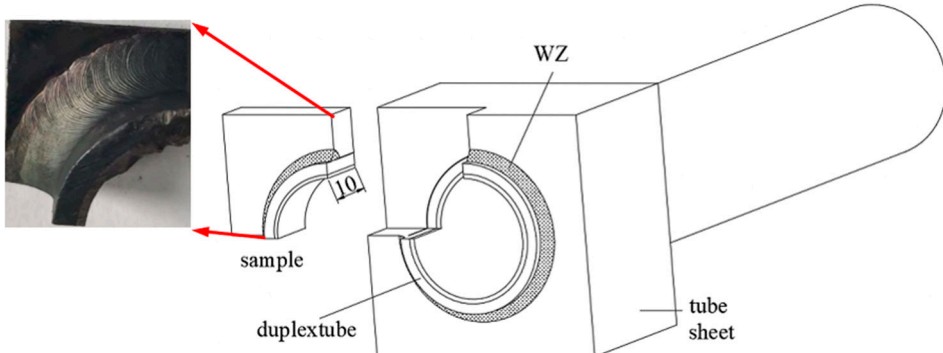

**Figure 2.** Schematic diagram and partial macro-drawing of welded sample.

The welded seam samples formed under different welding currents were placed in a three-electrode glass electrolytic cell to carry out electrochemical corrosion tests. The tested sample was connected to copper wire, covered with epoxy resin to expose only the surface of the weld joints with the area of 1 cm², as the working electrode. Ag/AgCl 3M KCl electrode was used as the reference electrode and platinum plate was acted as the account electrode, and all experiments were carried out in 3.5 wt.% NaCl solution by an electrochemical workstation (CHI 660E, Shanghai CH Instruments Co., Shanghai, China). The open circuit potential (OCP) was stabilized within 30 min with a scan rate of 10 mV/s before recording the polarization curves. The corrosion potential and corrosion current density of the weld joints were calculated by Tafel extrapolation. The electrochemical impedance spectroscopy (EIS) was carried out at the steady-state OCP potential value with the frequency range from $10^{-2}$ to $10^5$ Hz. The corrosion rate of the weld joint was calculated by the following formula (2):

$$CR = \frac{A \times I_{corr}}{n \times F \times \rho} \times 87600 \tag{2}$$

where $A$ is the atomic weight, $I_{corr}$ is the self-corrosion current density (A/cm²), $n$ is the number of electrons transferred by the electrochemical reaction, $F$ is the Faraday constant (C/mol), and $\rho$ is the metal density (g/cm³).

## 3. Results and Discussion

### 3.1. Surface Morphology

Figure 3 shows the SEM images of duplextube. It can be observed that the thickness of the Incoloy 825 alloy layer was about 640 µm. There is a transition layer between the alloy layer and low carbon steel. The thickness of the transition layer was about 50 µm. The interface between the alloy layer and carbon steel was metallurgical bonding mode by the pressure melting anchoring bimetal metallurgy technology. When the alloy tube was built in the heating base tube, the transition layer was formed under high pressure and high temperature during the hard-drawn process. However, there are some defects in the transition layer, such as micropores and voids. However, these defects in the transition layer were attributed to the existence of oxides inner surface of base tube and outer side of ally layer. These defects in the transition layer will influence the mechanical properties of the dupluxtube. The chemical composition of the alloy layer is shown in Figures 4 and 5 and Table 3. The Ti, Cr, and Ni elements were well distributed in the alloy layer (Figure 4). The nickel content in the alloy layer was high (about $41.3 \pm 1.23$ wt.%).

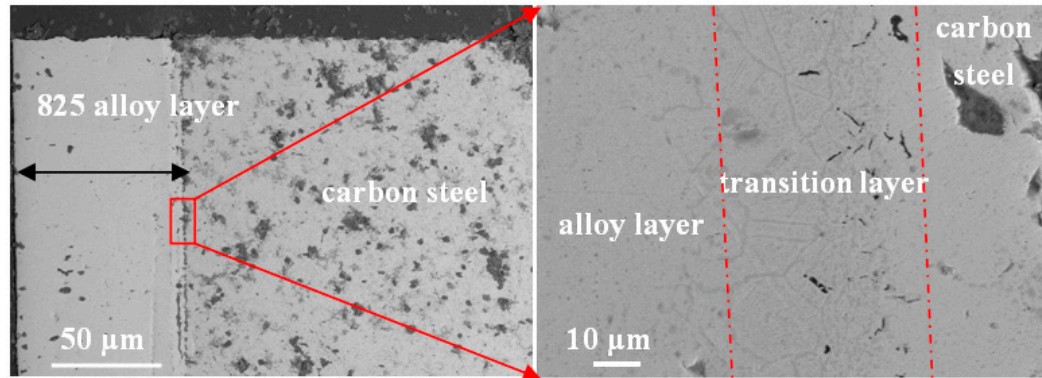

**Figure 3.** SEM images of duplextube.

**Table 3.** Chemical composition of Incoloy 825 alloy layer.

| Element | Mass % | Error % | At % |
| --- | --- | --- | --- |
| Ti | 0.73 | 0.3 | 0.85 |
| Cr | 23 | 0.48 | 24.76 |
| Fe | 34.96 | 0.68 | 35.03 |
| Ni | 41.3 | 1.23 | 39.36 |
| Total | 100 | | 100 |

Figure 6a shows a welded sample with welding current of 160 A; the black dot line is the base metal contour. It can be found that there was a clearance between the composite tube and the tube sheet (except for in the welding area). It can be proved that there was a strong fusion zone between the weld area and the base metal. All weld zones are continuous and have no obvious surface defects. This indicated that the heat input generated by welding is enough to completely melt the S10C steel of the tube, plate, and wire under three welding currents. It can be observed that the HAZ area between the dulpextube and ER50-6 filler materials did not influence the interface between the Incoloy 825 alloy layer and S10C steel. In Figure 6b, the weld zone thickness slightly increases with the increase of the welding current, while the weld width did not change. This is due to the fact thatwhen the welding current increases, the diameter of the arc column increases, and the depth of the arc penetrating into the workpiece increases. However, the moving space of the arc spot is limited, so the weld width does not change. When the welding current increases, the arc force and heat input on the workpiece increase, and the position of the heat source moves down, so the penetration increases. At the same time, the melting

amount of the welding wire increases in proportion, and the weld width is kept stable, so the weld zone thickness increases.

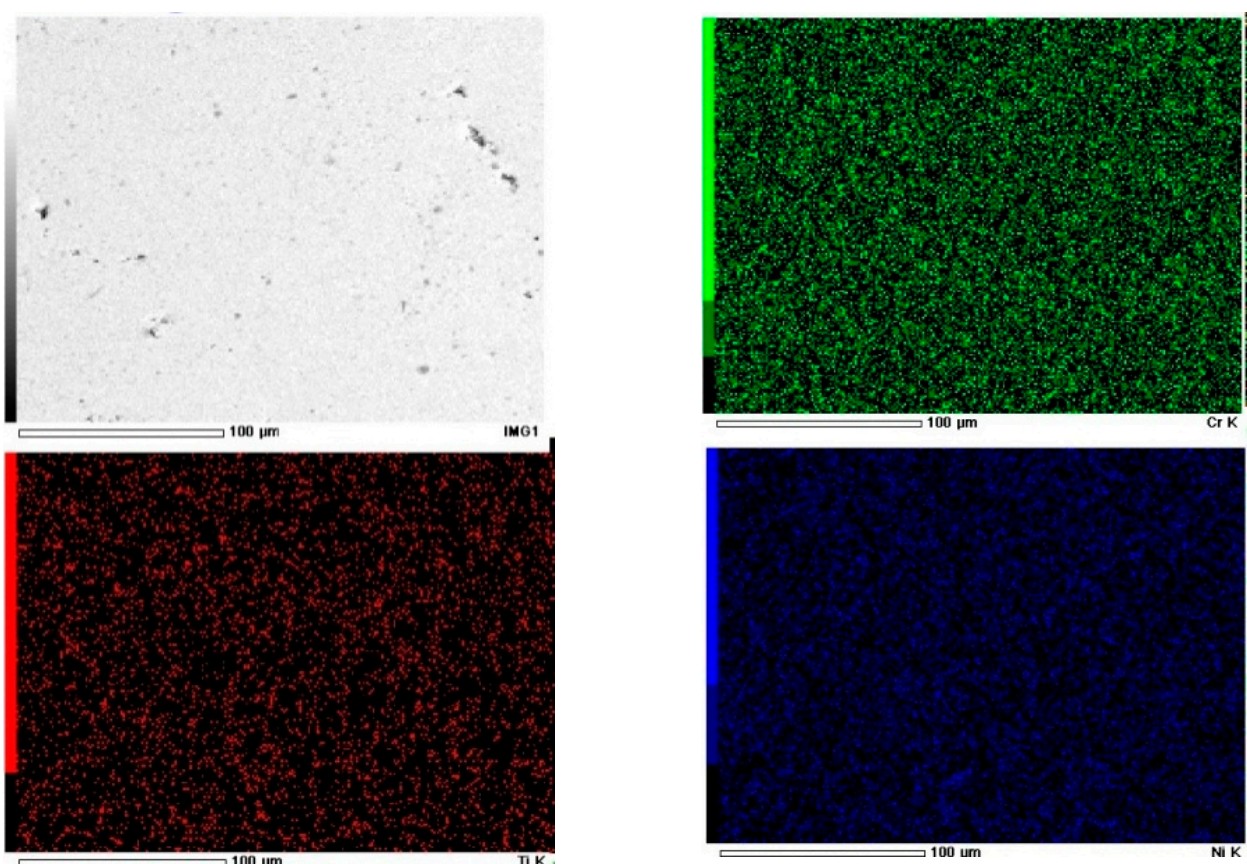

**Figure 4.** SEM image and surface scan energy maps of Incoloy 825 alloy layer (red color-Ti, blue color-Ni, and green color-Cr).

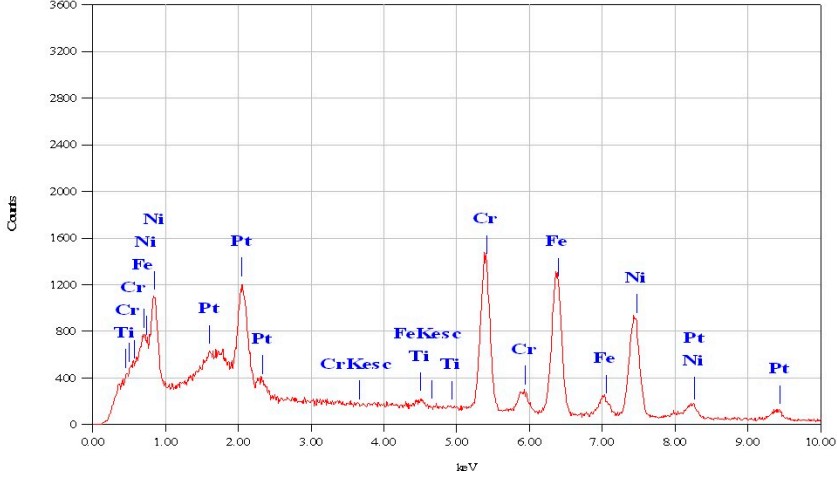

**Figure 5.** EDS pattern of Incoloy 825 alloy layer from Figure 4.

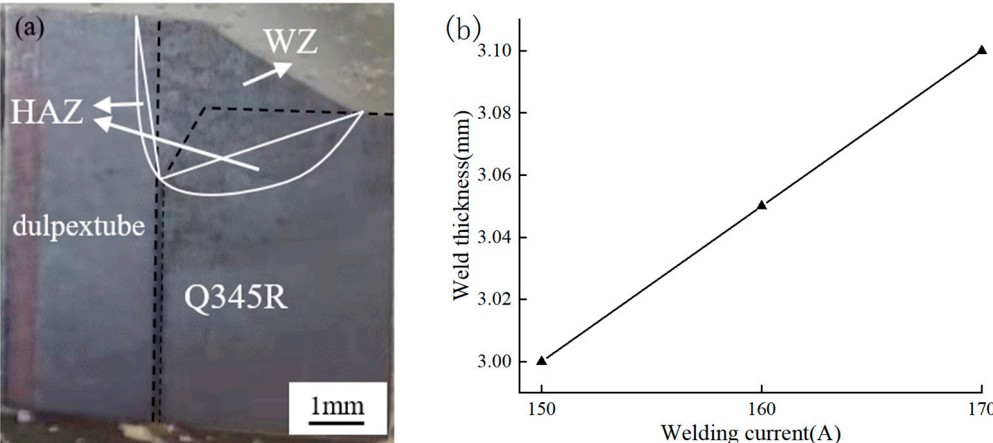

**Figure 6.** Welding section and welded width (**a**) weld joint section (**b**) weld width changed with welding current.

### 3.2. Metallurgical Characterization

Figure 7 shows the optical metallurgical micrographs of the weld and the base metal for sample #1. The S10C steel and Q345R is composed of α-Fe + pearlite phases with large grain size, compared with the metallurgical structure of the base material, the metallurgical structure of the weld area has undergone significant changes, and the phases of the weld zone (WZ) comprises δ−Fe and residual austenite. The fine grain size of the WZ is smaller than that of S10C steel and Q345R. During the welding process, the α−Fe in the base material becomes δ−Fe and austenite as the temperature increases, but when the fusion weld solidifies and cools to room temperature, δ−Fe and austenite still exist due to the process of primary crystallization. The equilibrium content of impurities such as S and P in the solid phase is also much lower than that of the liquid phase, resulting in the transfer of S and P in the base metal at the near interface to the weld. The impurity element enrichment layer is formed in the liquid phase at the front of the cross-section, and then rapidly solidifies, resulting in uneven chemical composition in the weld fusion area and rapid elemental segregation [15,16]. In summary, the input heat influenced the balance of ferrite and austenite, which could further the corrosion resistance of the weld joints.

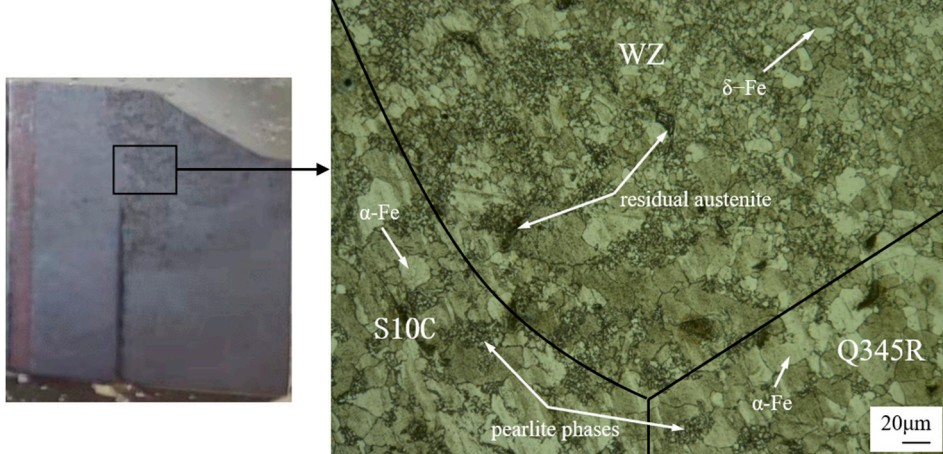

**Figure 7.** Digital camera photo and optical micrographs of weld zone (WZ) and base pipe for sample #1.

Figure 8 shows the optical metallographic photos of weld zone under different welding currents of 150 A, 160 A, and 170 A. It can be observed that that the grain size in Figure 8a is the smallest. As the welding current increased to 160 A, the grain size slightly increased (Figure 8b). Further, the grain remarkably increased when the welding current was equal

to 170 A (Figure 8c). It can be referred that the grain size also increases with the increase of the welding current, the other welding parameters are kept stable. The larger welding current, the higher the thermal energy per unit length of the weld, the greater the available energy, and at the same time, the weld cooling rate becomes slow due to the increase of the heat input (so that the grains were grown up).

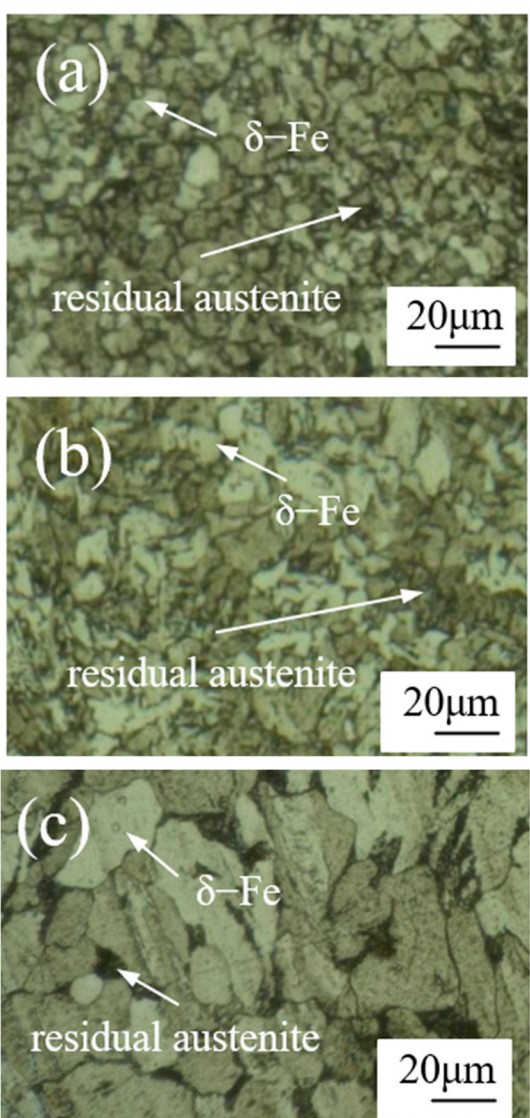

**Figure 8.** Optical micrographs of weld zone after welding process with different welding currents of (**a**) 150A for sample #1, (**b**) 160A for sample #2, and (**c**) 170A for sample #3.

### 3.3. Corrosion Resistance

Figure 9 shows the potentiodynamic polarization curves of three different weld joint samples. It can be seen that there are obvious differences in the polarization curves of the three samples, which is due to the different passivation ability of the samples during the corrosion test and to form a thin oxidation protective film [17]. According to Abioye's research [18], samples with high corrosion potential and low corrosion current density have better corrosion resistance than samples with low corrosion current density and high corrosion current density. The polarization resistance $R_p$ characterizes the corrosion resistance of the samples, and the larger the polarization resistance $R_p$, the better the corrosion resistance. Among the three weld joint samples, sample #3 welded with the largest welding current has the highest corrosion potential, the smallest corrosion current

density and the largest polarization resistance, while sample #1with the largest welding current has the lowest corrosion potential, the largest corrosion current density and the lowest polarization resistance, indicating that the corrosion resistance of sample #3 is the best. At the same time, according to the corrosion rate of the weld joint samples, the corrosion rate of sample#3 is the smallest, and the corrosion rate of sample #1 is the largest, it is shown that the corrosion resistance of sample #3 is better than that of samples #1 and #2. The electrochemical corrosion data is shown in Table 4. Compared with weld zone by welding wire-ER308L with high nickel content [19], the corrosion potential of the weld joints by ER50-6 weld wires was positive, but the corrosion current density is relatively high. Sincethe material component of weld zone was different, the corrosion resistance of the weld joint with high nickel and chromium elements is better than that of the weld joint with carbon steel. Above the corrosion potential ($E_{corr}$), the current density is in the order of sample #1 > sample #2 > sample #3. It can be inferred that the protective effect of passivation layer formed on the weld joint samples #1 and #2 is not good, compared with sample #3. However, the passivation region of sample #1 is the largest, and the pitting potential is also the highest. On the contrary, sample #3 has the narrow passivation region and the more negative pitting potential. The pitting corrosion potential values of sample #1, #2 and #3 are −62 mV, −101 mV, and −200 mV, respectively. Therefore, the pitting corrosion resistance of sample #1 is good, although the corrosion resistance of sample #1 is poor.

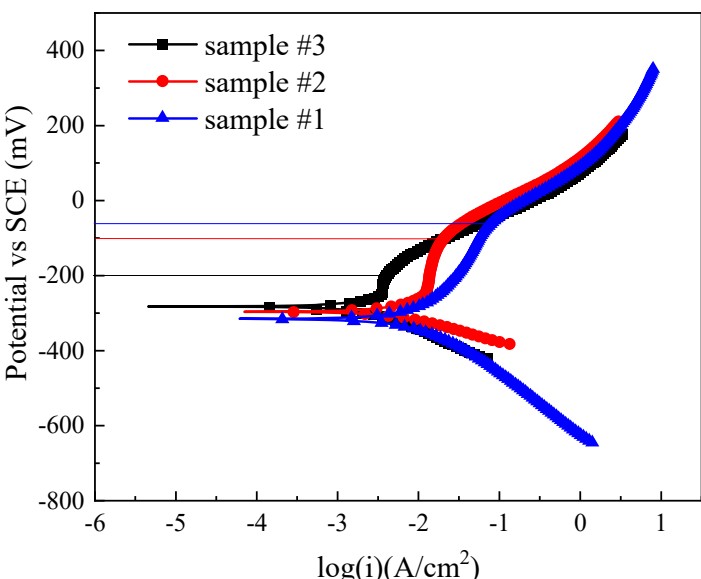

**Figure 9.** Polarization curves of three weld joint samples.

**Table 4.** Corrosive properties of the weld joints in 3.5 wt.% NaCl solution.

| Sample | $E_{corr}$ (mV) | $I_{corr}$ (mA/cm²) | $R_p$ (Ω/cm²) | CR (mm/a) |
|---|---|---|---|---|
| #1 | −307 | 0.093 | 28.36 | 1.47 |
| #2 | −297 | 0.065 | 42.61 | 1.02 |
| #3 | −286 | 0.041 | 54.87 | 0.64 |
| Heat input of 95J/cm * | −339 | 0.00571 | / | / |
| Heat input of 80J/cm * | −773 | 0.0132 | / | / |
| Heat input of 66J/cm * | −412 | 0.0115 | / | / |

Note: * Weld zone by welding wire-ER308L (Ni 12.83 wt.%, Cr 20.15 wt.%, C 0.028 wt.%, Si 0.60 wt.%, Mn 0.36 wt.%, P 0.004 wt.%, S 0.003 wt.%) [18].

Figure 10 shows the Nyquist plots and Bode diagrams of three samples in 3.5 wt.% NaCl solution. Figure 10a shows the capacitance impedance arc radius of three samples.

The larger the impedance radius, the stronger the corrosion resistance. Compared with samples #1 and #2, samples #3 has a larger impedance radius. Therefore, sample #3 with welding current of 170A has the highest impedance modulus and the best corrosion resistance. Ordinarily, the impedance value of low-frequency (0.01 Hz) could reflect the corrosion resistance of the sample. Moreover, a higher impedance value meant excellent corrosion protection. In the range from low frequency to medium frequency ($10^{-2}$ Hz), the impedance for sample #3 is the largest (Figure 10b), indicating the good corrosion resistance. Figure 10c shows that the phase angle of three samples. The phase angle firstly increases and then decreases with the increase of frequency and continues to decrease after reaching the extreme value. The phase angle value of sample #3 always was the highest (about $-70°$). The change trend of phase angle was the same, indicating the corrosion mechanism is the same. However, the phase angle and the impedance for sample #3 at the low frequency stage are the highest, sample #3 exhibited the best corrosion resistance, compared with the other samples. Due to the large grain size in sample #3, the number of grain boundaries was less than that of the other samples, resulting in good corrosion resistance.

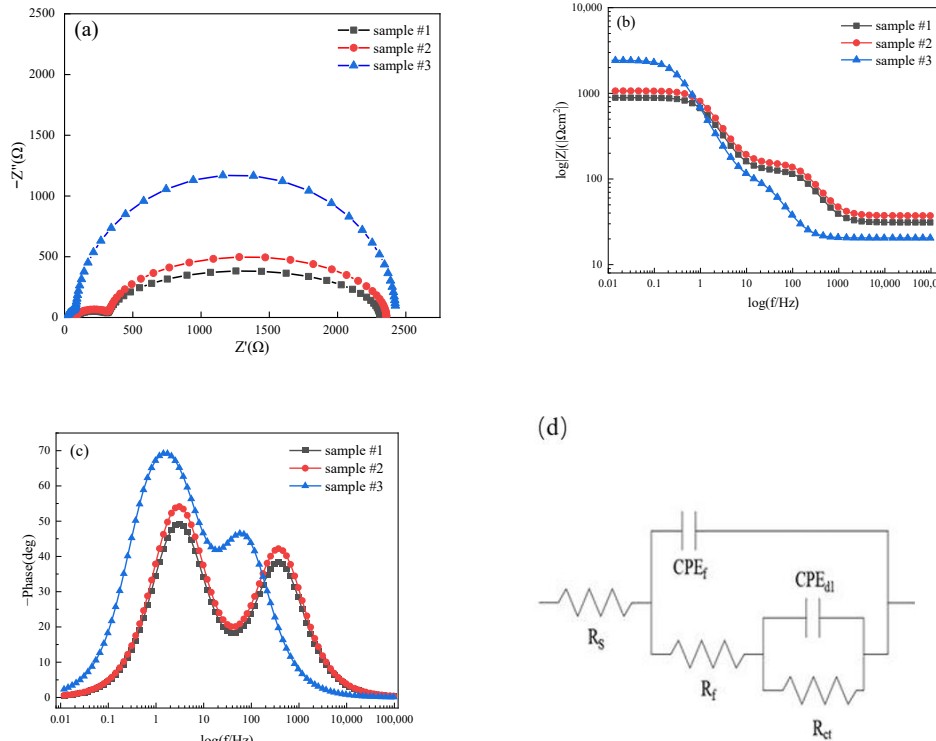

**Figure 10.** Electrochemical impedance spectra of the weld joints under different welding currents, (**a**) Nyquist plots, (**b**) impedance bode diagrams (**c**) phase angle, and (**d**) equivalent circuit diagram.

According to the EIS results of three samples, the equivalent circuit diagram is shown in Figure 10d. Due to the formation of passive film on the surface of three samples, the equivalent circuit is composed of two capacitance loops. $R_S$ and $R_f$ represent the solution resistance and passive film resistance, respectively. $R_{ct}$ is the electrolyte resistance. The defects of $CPF_f$ and $CPE_d$ are related to the passive film, respectively. The CPE can be expressed by formula (3) [20]:

$$Z_{CPE} = Y_0^{-1}(j\omega)^{-n} \tag{3}$$

where $n$ is its exponent, $\omega$ is the angular frequency, and $Y_0$ is the normal phase angle CPE constant. When $n = 0$, CPE presents resistance. As $n = 1$, CPE presents pure capacitance. Table 5 shows the results of EIS equivalent circuit. The larger $R_f$ value is, the greater the resistance of the oxide film is, and the stronger the corrosion resistance is. Therefore, the corrosion resistance of the weld joints is sorted in order of sample #3 > sample #2 > sample #1.

**Table 5.** Equivalent circuit parameters.

| Sample | #1 | #2 | #3 |
|---|---|---|---|
| $R_s$ ($\Omega \cdot cm^2$) | 31.09 | 26.73 | 20.49 |
| $CPE_f$ (F$\cdot cm^2$) | $8.48 \times 10^{-6}$ | $9.59 \times 10^{-5}$ | $5.66 \times 10^{-5}$ |
| $R_f$ ($\Omega \cdot cm^2$) | 101.7 | 104.7 | 106.4 |
| $CPE_d$ (F$\cdot cm^2$) | $1.81 \times 10^{-4}$ | $1.83 \times 10^{-4}$ | $1.86 \times 10^{-4}$ |
| $R_{ct}$ ($\Omega \cdot cm^2$) | 756.7 | 593.6 | 230.1 |

From what has been discussed above, the welding current affect the morphology, grain size and corrosion property of the weld zone significantly. Figure 11 illustrates the corrosion enhancement mechanism of fusion welding zone for duplextubes welded with Q345R tube sheet under different welding currents. The corrosion resistance mechanism of the weld zone by the welding current is that high welding current can improve the microstructure of the samples, which enhances the physical barrier property of the weld zone [21]. Besides, the fine structure creates more diffusion paths (see Figure 11c), and thus significantly improves the corrosion resistance of the weld zone. However, in this study, the results were opposite, which could be explained by the difference in the ferrite/austenite phase balance by different welding currents. On the other hand, the fine structure grains for samples #1 and #2 support much more corrosion sites (red dotted line) among the grain boundaries in the corrosion solution in the presence of chloride ion (see Figure 11a,b), compared with sample #3 with large grain size. Therefore, sample #3 had good corrosion resistance in 3.5 wt.% NaCl solution.

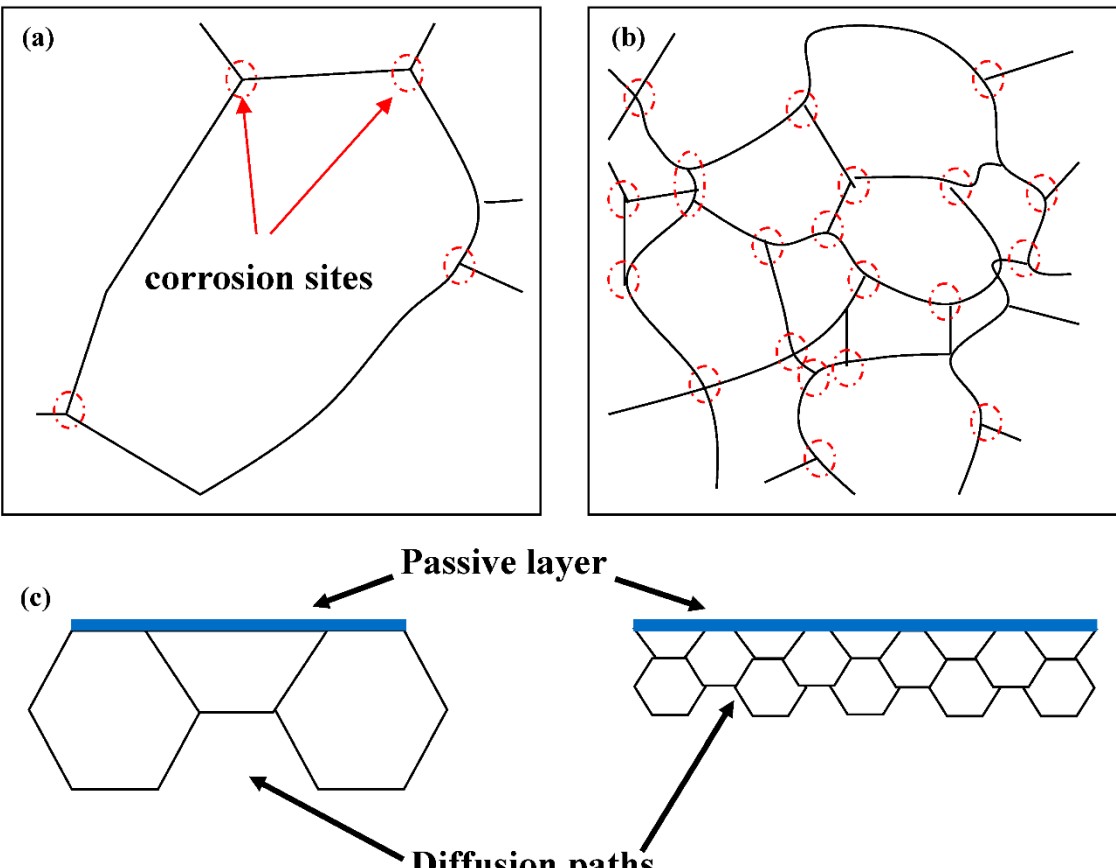

**Figure 11.** Schematic of the effect of grain size on corrosion properties of weld joint. (**a**) the surface of large grain, (**b**) the surface of small grain and (**c**) the cross-section of large and small grains.

## 4. Conclusions

In this study, S10C steel/Incoloy825 duplextubes and Q345R tube sheet was welded by gas tungsten gas tungsten arc welding technology with fillers ER50-6 carbon steel welding wire, the microstructure and corrosion behavior of fusion welding zone for duplextubes welded with Q345R tube sheet under different welding currents were studied. This work can provide a preliminary exploration for the manufacture of bimetallic composite tube air coolers. The main conclusions are summarized below:

(1) When the welding current is 150 A, 160 A, 170 A, the voltage is 14 V and the welding speed is 2 mm/s, the welded joints without obvious cracks and pores can be obtained, and the weld width increases slightly with the increase of welding current.

(2) The input heat influenced the balance of ferrite and austenite, which could further the performance of the weld joints. The phases of the weld zone are composed of $\delta-Fe$ and residual austenite phases. The weld joint has better grain refinement than the parent material, and the grain size of the weld joint increases with the increase of the welding current.

(3) Welding current has significant influence on the corrosion resistance of welded joints in 3.5 wt.% NaCl solution. The corrosion resistance of sample #3 welded with the highest welding current of 170A was better than that of samples #1 and #2. However, the pitting corrosion potential values of sample #1, #2, and #3 are $-62$ mV, $-101$ mV, and $-200$ mV, respectively. The pitting corrosion resistance of sample #1 with the lowest welding current of 150A is better than that of samples #2 and #3. The residual stress in the samples will be further investigated in future.

**Author Contributions:** Conceptualization, G.O.; methodology, G.O., H.J., G.Q. and W.W.; software, G.Q.; validation, G.O., G.Q. and H.J.; investigation, G.Q. and W.W.; resources, G.O. and H.J.; data curation, G.Q.; writing—original draft preparation, G.Q.; writing—review and editing, W.W. and H.J.; visualization, Q.L.; supervision, G.O.; project administration, G.O.; funding acquisition, G.O. All authors have read and agreed to the published version of the manuscript.

**Funding:** This work is supported by the National Natural Science Foundation of China (No. U1909216, No. 51876194, No. 52176048).

**Data Availability Statement:** Not applicated.

**Acknowledgments:** All authors thank Yaxuan Liu from Changzhou University to discuss the corrosion property of the weld joints.

**Conflicts of Interest:** The authors declare no conflict of interest.

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
