# Peer review of "Microstructure and Corrosion Resistance of Fusion Welding Zone for Duplextubes Welded with Q345R Tube Sheet under Different Welding Currents"

_metals, doi:10.3390/met12050705_

Round 1
Reviewer 1 Report
Dear authors, you must make a number of corrections before an article can be accepted.
- Figures 1 and 3 need to be corrected, since they have moved off the markings.
- It is necessary to add information about the equipment used (manufacturer, country).
- In the methods, you say that you used an optical microscope, but the results start with an analysis of SEM images. Complete the methods by indicating which microscope you used and in what modes.
- Figure 4 needs to be revised. In its current form it is very overloaded and does not provide useful information. Put element distribution maps in one figure, profiles in another, and add a table with EDS results.
- In Figures 6 and 7, remove the magnification, the dimensional marker is enough. Also add phase labels in the figures.
- Figure 8 should also be revised. As I understand, the figures were edited with microsoft word, so you should redo all the figures in a graphics editing software, as well as improve their quality, as some of the characters are simply not readable.
- The manuscript contains no discussion. The conclusions actually summarize the results, with only brief mention of corrosion, although the corrosion test results occupy most of the manuscript. Add a discussion and pay more attention to corrosion resistance in the conclusions.
Author Response
Reviewer #1
- Figures 1 and 3 need to be corrected, since they have moved off the markings.
Answer: We have corrected Figures 1 and 3.
- It is necessary to add information about the equipment used (manufacturer, country).
Answer: We have added information about the equipment used (manufacturer, country).
- In the methods, you say that you used an optical microscope, but the results start with an analysis of SEM images. Complete the methods by indicating which microscope you used and in what modes.
Answer: We have added some information about SEM and EDS.
- Figure 4 needs to be revised. In its current form it is very overloaded and does not provide useful information. Put element distribution maps in one figure, profiles in another, and add a table with EDS results.
Answer: Thanks! Figure 4 has been revised according to the comment.
- In Figures 6 and 7, remove the magnification, the dimensional marker is enough. Also add phase labels in the figures.
Answer: We have added dimensional makers and phase labels.
- Figure 8 should also be revised. As I understand, the figures were edited with microsoft word, so you should redo all the figures in a graphics editing software, as well as improve their quality, as some of the characters are simply not readable.
Answer: We have improved the quality of figure 8 with a graphics editing origin software.
- The manuscript contains no discussion. The conclusions actually summarize the results, with only brief mention of corrosion, although the corrosion test results occupy most of the manuscript. Add a discussion and pay more attention to corrosion resistance in the conclusions.
Answer: Thanks! We have added some discussion about pitting corrosion, and electrochemical corrosion mechanism.
Reviewer 2 Report
The manuscript describes the corrosion resistance of welding joints of tube sheets by varying the heat input. Two different materials were considered namely Steel and Q345R. It was concluded that weld current has a certain influence on the microstructure and corrosion resistance of welded joints: the size of the grain increases with the increase in the welding current and as a result the corrosion resistance increases. The conclusion was drawn based on experimental observation. The paper has merit. However, the following needs to be addressed before publication.
Abstract: It needs to be restructured to avoid confusion. For example, one might read that the grain size increases with the increase in welding current. But what about the 3.5 wt% NaCl solution? It needs to be stated clearly that the grain size you studied is just after the welding and then you studied the corrosion behavior by 3.5 wt% NaCl solution which shows better corrosion resistance. Also, 10# steel is not an international norm. Please use the international norm for this material.
Introduction: The reviewer is not convinced of the novelty of this manuscript. There is plenty of literature available on this topic. The reviewer suggests expanding the literature further to justify the need for this research.
Figure 1: The marks may have moved during the pdf conversion. Please make necessary corrections/precautions. Why two Weldings are appearing here? Where is the tube sheet? Why composite pipe appears in the figure without no mention of this material in the text?
Table 2: Please remove the speed and voltage columns as there were no changes on those columns and add that information in the table caption.
Please explain why 150A, 160A, and 170A were chosen. Is it a random choice or you follow a standard?
Figure 3: Texts are not clear. Please change the text colors. Please also include the scale bar.
Please include a scale bar in figure 5.
Improve the quality of figure 9.
Throughout the manuscript, it was mentioned as welding current. However, in the title, it says different heat input. I suggest changing either of them to be consistent.
Author Response
The manuscript describes the corrosion resistance of welding joints of tube sheets by varying the heat input. Two different materials were considered namely Steel and Q345R. It was concluded that weld current has a certain influence on the microstructure and corrosion resistance of welded joints: the size of the grain increases with the increase in the welding current and as a result the corrosion resistance increases. The conclusion was drawn based on experimental observation. The paper has merit. However, the following needs to be addressed before publication.
Abstract: It needs to be restructured to avoid confusion. For example, one might read that the grain size increases with the increase in welding current. But what about the 3.5 wt% NaCl solution? It needs to be stated clearly that the grain size you studied is just after the welding and then you studied the corrosion behavior by 3.5 wt% NaCl solution which shows better corrosion resistance. Also, 10# steel is not an international norm. Please use the international norm for this material.
Answer: Thanks! We have restructured the abstract and used the international norm S10C for 10# steel.
Introduction: The reviewer is not convinced of the novelty of this manuscript. There is plenty of literature available on this topic. The reviewer suggests expanding the literature further to justify the need for this research.
Answer: We have added some literature according to this comment and reviewer #3.
Figure 1: The marks may have moved during the pdf conversion. Please make necessary corrections/precautions. Why two Weldings are appearing here? Where is the tube sheet? Why composite pipe appears in the figure without no mention of this material in the text?
Answer: We have modified Figure 1. The reason for the two weldings is that I have done multiple welding experiments on the same tube sheet to simulate real air cooler welding conditions. I have markedthe tube sheet in Figure 1. The textual description of the composite pipe is in Experimental section.
Table 2: Please remove the speed and voltage columns as there were no changes on those columns and add that information in the table caption.
Answer: Thank you. We have changed the table 2.
Please explain why 150A, 160A, and 170A were chosen. Is it a random choice or you follow a standard?
Answer: The selection of welding current is determined by empirical value.
Figure 3: Texts are not clear. Please change the text colors. Please also include the scale bar.
Answer: Thanks. Figure 3 has been improved.
Please include a scale bar in figure 5.
Answer: We have added a scale bar in figure 5.
Improve the quality of figure 9.
Answer: The quality of figure 9 has been improved.
Throughout the manuscript, it was mentioned as welding current. However, in the title, it says different heat input. I suggest changing either of them to be consistent.
Answer: Thanks for the suggestion, the title has been improved.
Reviewer 3 Report
Review report on the topic ‘Microstructure and corrosion resistance of fusion welding zone for duplex tubes welded with Q345R tube sheet under different heat inputs’. Comments are listed below:
- Strengthen the abstract section. Remove the unnecessary information and add key points at the end of the abstract section.
- Discuss the novelty of the work in respect of the application.
- Try to make a bridge between current and previously published work and specify the gap area and objective of the work. Also, discuss the major process used for such type of joining and the problem associated with it. Refer to more papers and strengthen the introduction section: https://doi.org/10.1016/j.ijpvp.2021.104439; https://doi.org/10.1016/j.ijpvp.2021.104473.
- Provide the image of the experimental setup with good quality. Also, add the image of the welded pipe produced with proper scale.
- Check the caption of Fig. 1. There is no macrograph.
- Add the reference for equation 1 and also add the reference for efficiency: https://doi.org/10.1007/s13296-016-6007-z
- The quality of Fig. 3 is very poor. The interface of weld and base needs more discussion. Provide the discussion related to the transition zone, their formation and their role on mechanical properties. If possible, add the EDS line map or good quality area map for the interface region: https://doi.org/10.1007/s43452-021-00365-6; https://doi.org/10.1016/j.ijpvp.2021.104443.
- Instead of an optical image, provide a good quality SEM image. Also, the discussion of the interface is more important than the weld metal.
- The corrosion study is also presented in a rough manner.
- Improve the discussion quality throughout the manuscript.
Author Response
Review report on the topic ‘Microstructure and corrosion resistance of fusion welding zone for duplex tubes welded with Q345R tube sheet under different heat inputs’. Comments are listed below:
- Strengthen the abstract section. Remove the unnecessary information and add key points at the end of the abstract section.
Answer: We have revised the abstract section.
- Discuss the novelty of the work in respect of the application.
Answer: We have discussed the novelty of the work in respect of the application. "The high residual stresses, stress corrosion cracking, and metallurgical problems like forming an unmixed zone, δ−Fe formation in heat affected zone (HAZ), carbon migration, fusion zone, hydrogen-induced cracking occur in the weldment [12,13]. These problems must be evaluated before applicability in industries." has been added in the manuscript.
- Try to make a bridge between current and previously published work and specify the gap area and objective of the work. Also, discuss the major process used for such type of joining and the problem associated with it. Refer to more papers and strengthen the introduction section: https://doi.org/10.1016/j.ijpvp.2021.104439; https://doi.org/10.1016/j.ijpvp.2021.104473.
Answer: Some sentences have been added in the Introduction section.
- Provide the image of the experimental setup with good quality. Also, add the image of the welded pipe produced with proper scale.
Answer: These images have been added in the manuscript in the present best form.
- Check the caption of Fig. 1. There is no macrograph.
Answer: Yes, it is digital camera photo.
- Add the reference for equation 1 and also add the reference for efficiency: https://doi.org/10.1007/s13296-016-6007-z
Answer: We have added the reference for equation 1 and efficiency.
- The quality of Fig. 3 is very poor. The interface of weld and base needs more discussion. Provide the discussion related to the transition zone, their formation and their role on mechanical properties. If possible, add the EDS line map or good quality area map for the interface region: https://doi.org/10.1007/s43452-021-00365-6; https://doi.org/10.1016/j.ijpvp.2021.104443.
Answer: Figure 3 has been improved. The more discussion has been added about the interface of weld and base. However, the data about the EDS line map or good quality area map for the interface region could not be supported at present due to the boring pandemics.
"The interface between the alloy layer and carbon steel was metallurgical bonding mode by the pressure melting anchoring bimetal metallurgy technology. When the alloy tube was built in the heating base tube, the transition layer was formed under high pressure and high temperature during the hard-drawn process. However, there are some defects in the transition layer, such as micropores and voids. However, these defects in the transition layer were attributed to the existence of oxides inner surface of base tube and outer side of ally layer. These defects in transition layer will influence the mechanical properties of the dupluxtube." has been added in the manuscript.
- Instead of an optical image, provide a good quality SEM image. Also, the discussion of the interface is more important than the weld metal.
Answer: At present, this is the good micrograph. Due to the pandemic, the SEM images could not be done. The discussion of the interface has been added in the manuscript.
- The corrosion study is also presented in a rough manner.
Answer: We have improved the corrosion study.
- Improve the discussion quality throughout the manuscript.
Answer: We have improved the discussion quality throughout the manuscript.

Round 2
Reviewer 1 Report
Dear Authors, you revised your manuscript well in according with all comments and I will recommend it for publication.
Reviewer 2 Report
The authors made a significant effort to improve the manuscript by addressing the reviewer's comments. It can be accepted now.